# Post-Editing Neural MT in Medical LSP: Lexico-Grammatical Patterns and Distortion in the Communication of Specialized Knowledge

Hanna Martikainen

Université de Paris, CLILLAC-ARP, F-75013 Paris, France; hmarti@eila.univ-paris-diderot.fr

**Abstract:** The recent arrival on the market of high-performing neural MT engines will likely lead to a profound transformation of the translation profession. The purpose of this study is to explore how this paradigm change impacts the post-editing process, with a focus on lexico-grammatical patterns that are used in the communication of specialized knowledge. A corpus of 109 medical abstracts pre-translated from English into French by the neural MT engine DeepL and post-edited by master's students in translation was used to study potential distortions in the translation of lexico-grammatical patterns. The results suggest that neural MT leads to specific sources of distortion in the translation of these patterns, not unlike what has previously been observed in human translation. These observations highlight the need to pay particular attention to lexico-grammatical patterns when post-editing neural MT in order to achieve functional equivalence in the translation of specialized texts.

**Keywords:** neural MT; post-editing; functional equivalence; distortion in translation; lexico-grammatical patterns; medical LSP

## 1. Introduction

The past few years have witnessed the arrival on the market of neural machine translation (NMT) engines that appear extremely promising as their output convincingly mimics the product of human translation. This has spurred questions in the professional translation sector regarding the future of the profession and specifically the impact that NMT will have on the post-editing process. This paper investigates the specificities of NMT in the English-to-French translation of lexico-grammatical patterns in the medical domain, the potential for distortion therein, and thus, the requirements for post-editing NMT in a specialized domain.

### 1.1. Post-Editing Neural Machine Translation

The growing body of evidence on NMT suggests, globally, improvements in quality in comparison with the previous generation of statistical engines, but mixed results specifically in terms of post-editing effort. In specific language pairs and domains, NMT appears to result in significative improvements over statistical MT [1] (p. 110). For instance, it has been estimated through human evaluation that errors are reduced by 60% between the statistical and neural versions of Google's MT engine in the English–French and English–German language pairs [2] (p. 88). Similarly, in the English–Japanese language pair, a recent study concluded that texts pre-translated by Google's neural engine and post-edited by learners were globally of better quality and contained fewer errors than texts pre-translated by the statistical version of the same engine [2] (p. 102). Yet, in spite of similar perceived cognitive effort of post-editing MT output by statistical and neural engines, student post-editors corrected fewer errors in NMT output. The author suggests this could be because NMT results in errors that are similar to those produced by humans, which makes them more difficult to detect and therefore to post-edit [2] (p. 102).

It has indeed been shown that different types of MT engines produce different types of errors [3], and that increased fluency in NMT output comparatively to statistical engines is often obtained at the cost of accuracy [4,5]. Specifically, although automatic metrics on NMT are very promising, human evaluation has highlighted incoherent results in terms of accuracy, with more omissions, additions, and mistranslations, as well as higher post-editing effort [1] (p. 118). Thus, recent studies stress the need to train students specifically in post-editing of NMT [2,5], not only because language service providers increasingly require those skills [5], but also because the task of post-editing NMT would seem to require competences that are similar to conventional translation or revision [2].

In addition, some recent studies analyze the differences between post-editing NMT and conventional translation [6,7]. In the English–Chinese language pair, post-editing NMT appears significantly faster than translation from scratch on specialized texts, and also seems to reduce cognitive effort [6] (pp. 67–71). Although translations produced by the two processes were judged of equal quality in terms of accuracy and fluency, when asked to pick the best translation between different versions, human evaluators expressed a clear preference for human-translated sentences, which could be explained by stylistic issues [6] (pp. 75–77). In another study comparing the quality of translated and post-edited texts from the end-user perspective through eye-tracking and human evaluation in the English–Welsh language pair, texts produced by the two processes obtained similar results, which for the author suggests the absence of any difference in quality for end-user in terms of readability and comprehensibility [7] (p. 147).

### 1.2. Distortion in Specialized Translation

The present paper draws largely on research conducted for an ongoing PhD project at the Université de Paris [8], the main focus of which is distortion in specialized translation. The basic premise of this research is that translated texts are likely to contain elements, such as translation errors, that can potentially impact readers' interpretation of the message. The PhD project has a threefold objective: to categorize such potential sources of distortion; to establish their distribution in translated texts, specifically with a view of comparing conventional human translation and post-edited machine translation in terms of distortions; and to explore the impact of distortion on readers' interpretation. The project deals with a specific sub-genre within the medical domain, that of the systematic review abstract. Cochrane systematic reviews, which summarize existing data on a given medical question, are considered the gold-standard in this field. The reviews themselves are published in subscription databases but their abstracts are freely available online and translated by regional Cochrane centers from English into various languages, French among others. Cochrane reviews aim for a factual and objective presentation of research results in order to facilitate their transfer into medical practice. They can thus be considered to represent, according to Reiss's classification [9], a purely informative text type without any persuasive objective. In this context, then, distortion is defined as interference with the communicative purpose of the text. This interference can be due to different kinds of elements in the translated text that impact the interpretation of essential characteristics of the systematic review, for instance the effectiveness of the intervention or the authors' level of confidence in their results.

The first step in the PhD project consisted in categorizing such potential sources of distortion in a typology. This process, described in detail in [10], was essentially based on analysis of an existing collection of human-translated Cochrane reviews that had been previously reviewed for errors by domain experts from Cochrane France. Domain specialists' vision of what constitutes distortion in translation was thus accessed via their extensive comments on errors observed in translated texts. This approach was then refined through researches on a large corpus of translated Cochrane abstracts for fine-tuning the sub-categories in the typology of sources of distortion. Two kinds of elements were determined to have the potential to distort readers' interpretation, i.e., translation errors proper but also, biased translation of typical structures of specialized language called lexico-grammatical patterns, discussed in detail below. As readers' interpretation plays an important part in the perception of biased

translation, the potential impact of these distortions was subsequently confirmed by an online survey, also discussed further below.

### 1.3. Lexico-Grammatical Patterns as a Source of Distortion in Translation

The focus of this paper is on the translation of lexico-grammatical patterns, i.e., "the basic building blocks" of languages for specific purposes (LSPs) [11] (p. 75). Composed of stable pivotal elements and a more variable albeit regular paradigm, these "chains of meaningful interlocking lexical and grammatical structures" can be extended and are non-linear, and have regular textual functions [11] (p. 79). An example of a typical lexico-grammatical pattern in technical language that functions as a warning would be "FAILURE TO + V [= comply with, follow, observe, etc.] + MAY/CAN + LEAD TO/RESULT IN + N [= death, fire, injury, property damage, etc.]" [11] (p. 78). Within the ongoing PhD project [8], various lexico-grammatical patterns used in Cochrane review abstracts for the communication of specialized medical knowledge have been identified. These patterns fall into two main categories. Positive patterns have the communicative function of signaling potential effectiveness of interventions, for instance: "N [= Intervention under study] + MODAL AUX [= may/can/etc.] + V [= lead to a desired outcome, e.g., reduce mortality]". The function of negative patterns is to signal lack of evidence of effectiveness, for instance "We + EVIDENTIAL V [= find/show/etc.] + NEG + EVIDENCE OF + N [= benefit, effect]". Lexico-grammatical patterns identified in medical language often also contain modal markers, used specifically for signaling the degree of certainty of the results [12].

A functionally equivalent translation [9] of such patterns should imply similar levels of (un)certainty regarding the potential effectiveness of interventions—or lack thereof. Translated texts, however, generally tend to follow and even exaggerate target-language norms and conventions [13] which, in French-language scientific writing, call for markedly more affirmative and less hedged claims than is the case in English [14]. In addition, the modal markers often embedded in these patterns pose the problem of conferring mitigated claims [15] and have been shown to be a frequent source of uncertainty in medical translation [16]. Moreover, in medical LSP, hedging devices are mostly used as "real hedges", in that they are used "to convey real uncertainty, for example when the nature of the research findings does not allow the author to make strong claims or draw clear conclusions", in other words, "to be precise" [17] (p. 81). Indeed, it is common in the medical domain to "make claims mainly in a tentative, reserved and objective way" [18] (p. 1). Thus, translating such lexico-grammatical patterns from English into French in accordance with target-language norms and scientific writing conventions is likely to result in a shift in the level of certainty that interferes with the essential communicative function of these texts, i.e., distortion. As these more affirmative translations are culturally motivated, they can be considered biased in the sense of systematic distortions, as opposed to translation errors which are in essence random [10].

The expert reviewers' comments mentioned above were highly valuable for understanding how these biased translations of lexico-grammatical structures could potentially distort readers' interpretation. Indeed, domain experts were quick to comment on "added or subjective terms that can lead to misinterpretation" (Original comment in French: '*des termes ajoutés ou subjectifs qui peuvent mener à une mauvaise interprétation*') and frequently stressed in their comments the importance of these structures for communicating specialized knowledge, i.e., "In my view, this translation is not rigorous enough, whereas precision is essential in reporting results from meta-analysis" (Original comment in French: '*Je trouve que cette traduction n'est pas assez rigoureuse alors qu'il faut beaucoup de précision pour reporter les résultats d'une méta-analyse*'). It appears, however, that the norms of adequate and functional translation in specialized languages can be different from those in the general language: for instance, expert reviewers frequently opted for word-to-word calque when correcting the translation of existential framing structures (i.e., *There is evidence . . .* ).

### 1.4. Potential for Distortion in the Translation of Positive and Negative Pattens

For positive lexico-grammatical patterns signaling potential effectiveness of interventions, the distortion in translation arises specifically from more affirmative translation choices for certain markers occurring within these patterns, mainly modal auxiliaries and evidential verbs, a discussion of which can be found in [19]. For example, in the context of a positive lexico-grammatical pattern where potential effectiveness is signaled through the modal auxiliary *may* (i.e., *intervention may be effective*), a typical biased translation in French consists in using the verb *pouvoir* in the indicative mood (*peut*), expressing inherent potential, instead of the more hypothetical conditional mood (*pourrait*) [19]. The potential for positive bias due to the use of the indicative form in this context was also highlighted by some of the domain experts having reviewed previous translations of Cochrane abstracts, one of whom even stated that the indicative mood was "never" an adequate translation for the auxiliary *may*(Original comment in French: '*pourrait et jamais peut*'.).

Other markers identified as potentially subject to biased translation within positive lexico-grammatical patterns are the evidential verbs *show* and *find*. A functionally equivalent translation choice for both would be for instance the verb *montrer*, defined as "to permit to be seen" and synonymous with *indiquer (indicate)* [20] (see also [21] (p. 17) for a discussion of the contextual equivalence of *show* and *montrer* for reporting results in the biomedical domain). Typical translations that would be considered more affirmative and therefore positively biased in this context are the verbs *démontrer* and *révéler*. The first is defined as "to prove by rigorous reasoning", the second as "to make evident by undisputed signs", while both are synonymous with *prouver (prove)* [20] (see also [21] (p. 16) for a discussion of *démontrer* in reporting empirical demonstrations of evidence in the biomedical domain).

In the translation of negative patterns, the distortion is more specifically related to potential confusion between absence of evidence of effect and evidence of absence of effect. This is a common mistake that readers of Cochrane abstracts are frequently guarded against, as illustrated by the following excerpt from the corpus compiled for the present study:

> There was also no evidence that the studies reduced the risk of cardiovascular and metabolic disorders in childhood cancer survivors, *although no evidence of effect is not the same as evidence of no effect.*

A functional translation of such negative patterns would use the basic negative form (*ne . . . pas*), while a typical biased translation consists in using a stronger form of negation such as the adjective *aucun(e)* [10]. Again, the potential for negative bias due to the use of a stronger negative form was also highlighted by several of the domain experts having reviewed previous translations of Cochrane abstracts, who considered it to be a "mistranslation" (Original comment in French: '*faute de sens dans la traduction*'), i.e., "The translation is not precise enough for reporting of results: 'does not show a difference' and 'shows no difference' are not the same" (Original comment in French: '*Cette traduc[t]ion prend trop de liberté pour le report des résultats: on ne montre pas de différence n'est pas la même chose qu'on ne montre aucune différence*').

### 1.5. Perception of Distortion Due to Biased Translation

The interpretation of lexico-grammatical patterns and the perception of bias in translation are, however, subject to important individual variation and it was therefore necessary, within the framework of the ongoing PhD project [8], to confirm the potential impact of distortion for French-language readers. An online survey was conducted in June–July 2018 and taken by a total of 147 respondents. Each respondent evaluated the translation of one positive and one negative pattern randomly chosen from a pool of 5 positive and 4 negative examples. Respondents were asked to rate on a 5-point scale the equivalence, in terms of the degree of affirmativeness, of two French-language examples, one of which was the original biased translation and the other a functionally equivalent version of the same source text. Results of the survey suggest that the majority of respondents were receptive to the subtle nuances in the expression of uncertainty that are in play in lexico-grammatical distortion: globally, 69% of respondents found the original biased translations more affirmative than their functionally equivalent

versions. However, results also confirm that interpretation plays a major part in the perception of lexico-grammatical structures. Indeed, 12% of respondents considered the biased and neutral versions to be equivalent in terms of affirmativeness, whereas for another 9% of respondents, the functionally equivalent translation actually appeared more affirmative than the original biased translation. When broken down by type of pattern, survey results suggest more agreement in the perception of positive bias, with 91% of respondents considering the biased and neutral versions not equivalent in terms of their degree of affirmativeness, versus 85% of respondents for negative patterns.

### 1.6. Lexico-Grammatical Distortion in Translations Produced by Different Methods

Finally, one of the objectives of the PhD project [8] is to compare conventional human translation and post-edited statistical machine translation in terms of distortions. To this end, the typology of sources of distortion was used to map distribution of distortions in corpus. The results suggest that lexico-grammatical sources of distortion are more frequent in human-translated texts (51% of observed distortions) than in post-edited statistical MT (37% of observed distortions). Moreover, human translation is characterized by globally more affirmative translation choices for lexico-grammatical patterns comparatively to post-edited statistical MT. The results of this previous research will be referred to for comparison purposes when discussing the findings of the present study.

The main aim of the present study, then, was to detect potential tendencies observable in the translation of lexico-grammatical patterns by neural MT. On the basis of previous research [8], human-like tendencies—i.e., more affirmative translation solutions—were expected to be observed. The results of the present study suggest that NMT indeed convincingly mimics human translation, producing in some instances markedly more affirmative, human-like output for the lexico-grammatical patterns investigated. Moreover, it appears that student post-editors mostly tend to accept biased MT output as such. This highlights the need to pay attention to lexico-grammatical patterns when post-editing LSP, and to train translation students on these issues.

## 2. Materials and Methods

A corpus study of selected lexico-grammatical patterns was conducted on Cochrane systematic review abstracts pre-translated from English into French by the neural MT engine DeepL and post-edited by second-year master's students in specialized translation at the Université de Paris. The master's in Language Engineering and Specialized Translation (ILTS) is a professionally-oriented master's program within the European Masters in Translation (EMT) network for training future language specialists with skills adapted to market needs. Student selection for the second-year master's program is based on testing of their language and translation skills. Students admitted to the program must have perfect command of the French language, as well as excellent comprehension and writing skills in English. The practice of a third language is encouraged in addition to the English–French language pair which is extensively used for most courses within the program. Thus, although the students come from different backgrounds and have varied profiles in terms of previous education, work experience, age range or even their first language, all have to demonstrate the required competences in order to be admitted to the program. In the academic year 2018–2019, a total of 38 students were enrolled in the program, 8 of whom were non-native speakers of French.

### 2.1. Production Context

During the course of a six-month class, each student post-edited an average of three different texts and revised three texts post-edited by other students. The texts then went through a final revision by the teacher, also the author of the present paper, before they were published on the different Cochrane websites. Although no specific translation brief was provided in this actual production context, students were instructed to take their time to ensure a factually accurate translation and to avoid preferential stylistic changes that do not improve fluency. Students were introduced to a collection of specific resources for medical translation and encouraged to test them in their work so as to determine

which best suited their individual preferences. The texts were post-edited via the online platform Memsource on which DeepL was integrated. Although the students were familiar with comparable interfaces and some had been previously introduced to Memsource within another class, most were novices to both post-editing and medical translation at the beginning of the class. Experience in both was acquired over the duration of the class, and theoretical teachings and practical exercises were used in support of the post-editing work done by students. The importance of lexico-grammatical patterns in specialized languages and the potential for bias in their translation were specifically discussed approximately half-way through the course.

The fact that the study corpus is derived from an actual production context means that various parameters that may have influenced the results and their interpretation could not be controlled. This is often the case for corpus studies, which rely on observations drawn from real-life data instead of strictly limited experiments with a controlled set of parameters. The potential impact of these limitations on the results of the present study is discussed in Section 4.

### 2.2. Corpus Characteristics

Data was collected directly from the online platform Memsource in MXLIFF and TMX formats and converted to Excel for uploading to Sketch Engine.The study corpus is available upon request in MS Excel format. The corpus was automatically tagged with TreeTagger when uploaded onto Sketch Engine. The corpus consists of 109 different texts (Cochrane abstracts and plain language summaries) in three versions: English source texts, corresponding NMT output in French, and initial student post-editions of NMT output. The three sub-corpora are aligned on sentence level. Table 1 presents token counts for each sub-corpus as well as the translation coefficient, i.e., the token count ratio, for the two French-language sub-corpora. The coefficient is somewhat higher than the usually quoted expansion rate for this language pair, i.e., 15–20% [22] (p. 159), but similar to what was previously observed for these texts [19].

**Table 1.** Study corpus token count and translation coefficient.

| Sub-Corpus | English | French NMT | French Post-Edited |
|---|---|---|---|
| Token count | 125,865 | 162,463 | 165,516 |
| Translation coefficient | - | 1.29 | 1.32 (1.02) [1] |

[1] Translation coefficients from English and NMT output, respectively.

### 2.3. Lexico-Grammatical Patterns Studied

The following lexico-grammatical patterns and specific markers thereinwere selected for study on the basis of previous work [8], as being potentially subject to biased translation. (Abbreviations used in the patterns: ADJ = Adjective; AUX = Modal Auxiliary Verb; EV = Evidential Verb; N = Noun; NEG = Negation; V = Verb. Optional elements are presented between parentheses.)

- Positive patterns signaling potential effectiveness built around evidential verbs

  (a)  N [= Intervention] + EV [= seem/appear to] + V [= have a beneficial effect]
  (b)  N [= Intervention/Results/etc.] + EV [= show/suggest/etc.] + N [= effectiveness/benefit/etc.]

- Positive patterns signaling potential effectiveness built around modal auxiliary verbs

  (c)  N [= Intervention] + AUX [= may/can/etc.] + V [= lead to a desired outcome]

- Framing structures for positive lexico-grammatical patterns. Positive patterns can either be standalone or associated with a framing structure to form extended patterns.

  (d)  We/N [= Study/Trial/etc.] + EV [= find/show/etc.] + N [= evidence] + OF/THAT

(e)     THERE + BE + N [= evidence] + OF/THAT

- Negative patterns signaling lack of evidence of effect

(f)     FRAME [= There was/Results show/etc.] + NEG + (ADJ) + N [= effect/difference/etc.]

(g)     NEG + (ADJ) + N [= effect/difference/etc.] + BE + EV [= find/show/etc.]

As discussed in Section 1.3, the functions of positive lexico-grammatical patterns are to express potential effectiveness of interventions and to grade the authors' confidence in the results. They have a high number of variable lexical parameters (i.e., interventions, criteria of effectiveness) and are therefore not easily constrained for searching in corpus. These patterns are, however, built around specific modal markers, some of which were previously observed to be subject to biased translation [19]. These markers were used as anchor points in searching for positive patterns. First, lists of verbs, modal auxiliaries, and adjectives/adverbs in the English source text sub-corpus were established through Sketch Engine, and then a manual selection of evidential verbs, epistemic modal auxiliaries, and modal adjectives was operated from the respective lists. Corpus concordances for the selected markers were then screened for instances of use within positive lexico-grammatical patterns related to treatment effectiveness and their contexts examined for additional evaluative or modal markers.

Negative patterns that serve the function of expressing lack of evidence of effect (see Section 1.3) are more straightforward to constrain for searching in corpus as they are built around specific markers, i.e., the negative marker NO(T) associated with a lexical marker such as *evidence*, *effect*, *difference* or *benefit*. Concordances resulting from the searches were manually checked and their contexts examined for additional modal markers such as adjectives or adverbs and evidential verbs.

For both the positive and negative patterns identified in corpus, MT output as well as the post-edited versions were then examined for instances of distortion. For certain specific markers (i.e., occurring within patterns a–c or f–g above), the corpus was searched for translation options that had previously been determined as being biased, such as the negative marker *aucun* or the indicative form of the verb *pouvoir*. For other patterns (mainly, d and e), concordances were examined manually to detect potentially biased versions among the translations.

## 3. Results

### 3.1. Positive Lexico-Grammaticcal Patterns Signaling Potential Effectiveness of Interventions

Positive patterns presented in this subsection are grouped according to the marker mainly subject to biased translation, i.e., an evidential verb or a modal auxiliary. Modal clusters as well as framing structures are examined separately.

3.1.1. Patterns Built around Evidential Verbs

The corpus was first searched for all tokens with a POS-tag corresponding to a verb form ("VV.*"). A total of 12,647 tokens were retrieved by the search, corresponding to 1449 different word forms. The list of verb word forms was then screened for evidential verbs, and 14 potential candidates were found (*appear, confirm, demonstrate, favour, find, indicate, note, observe, reveal, see, seem, show, suggest, support*). After exclusion of negative patterns through specific markers, results of corpus searches were manually refined in order to retain only occurrences of these evidential verbs within positive lexico-grammatical patterns signaling potential beneficial effects of interventions (see Section 2, patterns a, b, and d). Verbs occurring less than five times within such a pattern were excluded from further study (7 in total). For the remaining seven evidential verbs, aligned concordances were examined for occurrences of biased machine translation, and the corresponding post-edited versions were checked for modifications. Results are presented in Table 2.

**Table 2.** Evidential verbs subject to biased machine translation within positive patterns.

| Marker | Total Occurrences | Within Positive Patterns | >Biased MT | >Biased PE |
|---|---|---|---|---|
| appear | 43 | 19 | 0 | 0 |
| demonstrate | 18 | 6 | 1 | 1 |
| favour | 14 | 13 | 0 | 0 |
| find | 327 | 55 | 13 | 13 |
| indicate | 24 | 8 | 0 | 0 |
| show | 146 | 55 | 12 | 11 |
| suggest | 76 | 31 | 1 | 1 |

Most of these evidential verbs are machine-translated by their closest equivalents in the target language which presumably correspond to similar levels of (un)certainty, such as *sembler* for *appear*, or *suggérer* for *suggest*, and thus can be considered functionally equivalent translations. Two evidential verbs, *find* and *show*, which both represent factual and neutral choices in the source language, are however more frequently subject to shifts in MT output towards more affirmative, biased lexical choices implying a higher level of proof, as discussed in Section 1.4. In the case of *find*, neutral translation options are represented for instance by the verbs *constater* or *trouver*, whereas the typical biased translation is *révéler* (see Example 1 below). In MT output, 13/55 (~24%) occurrences of *find* in the context of positive lexico-grammatical patterns are subject to biased translation. (For all the examples presented here, the contextual window necessary for the MT engine to produce the output that appeared in the actual production context is quoted in total, since NMT is subject to important variation with even minor changes to the segment. A simplified back-translation is given to explain the origin of the bias.)

**Example 1.** Biased translation of *find* within a positive pattern.

> One trial <u>found</u> large benefits for SBP and DBP (SBP . . . ).
> MT: Un essai <u>a révélé</u> des avantages importants pour la SBP et la DBP (SBP . . . ).
> [= trial REVEALED important advantages for outcomes]

As discussed in Section 1.4., for the evidential verb *show*, a neutral and functionally equivalent translation choice is *montrer*, while the typical biased translation implying a greater degree of certainty is *démontrer* (see Example 2 below). In the corpus sample, *show* is subject to biased machine translation in 12/55 (~22%) of its occurrences in the context of a positive lexico-grammatical pattern signaling potential benefits of interventions.

**Example 2.** Biased translation of *show* within a positive pattern.

> Data on dropouts from two studies with 95 participants <u>showed</u> a clear advantage for couple therapy (RR 0.31 . . . ).
> MT: Les données sur les abandons de deux études menées auprès de 95 participants ont <u>démontré</u> un net avantage pour la thérapie de couple (RR 0,31 . . . ).
> [= data DEMONSTRATED/PROVED a distinct advantage for the intervention]

The three biased translation choices (*démontrer, (se) révéler, s'avérer*) observed in the context of positive patterns represent approximately 19% (28/146) of total occurrences of *show* in the corpus sample. For comparison purposes, on a larger corpus of Cochrane abstracts compiled for the aforementioned PhD project [8], those three translation choices represent 25% of total occurrences of *show* in human-translated texts, versus 12% in statistical MT post-edited by professionals. This would seem to suggest that, in terms of evidential modal verbs, neural MT output tends more towards human translation than to statistical MT.

Finally, as can be seen in Table 2, it appears that student post-editors tended to accept the formulations suggested by the MT engine as such without any modifications. Indeed, for evidential verbs within positive lexico-grammatical patterns, the biased translation suggested by DeepL was edited in only one single instance, where the verb *show* pre-translated by *démontrer* was replaced in the post-editing stage by the more neutral *montrer*.

### 3.1.2. Patterns Built around Modal Auxiliaries

A total of 894 occurrences of modal auxiliaries were retrieved by the search (POS-tag "MD") and correspond to eight different modal verbs (in order of frequency in the corpus sample, *may, can, could, should, might, would, will, must*). Four of them mainly occur in epistemic use in the context of the positive lexico-grammatical pattern under study related to effectiveness of interventions (see Section 2, pattern c), wherein their function is to signal different degrees of certainty. The total frequencies of these auxiliary verbs in the corpus sample, occurrences within positive lexico-grammatical patterns, and occurrences of biased translation among the latter are presented in Table 3.

**Table 3.** Modal auxiliaries subject to biased machine translation within positive patterns.

| Marker | Total Occurrences | Within Positive Patterns | >Biased MT | >Biased PE |
|--------|-------------------|--------------------------|------------|------------|
| can | 235 | 23 | 0 | 0 |
| could | 96 | 16 | 0 | 0 |
| may | 391 | 155 | 142 | 135 |
| might | 41 | 20 | 3 | 3 |

The auxiliaries *can* and *could* are used respectively within these patterns to signal capacity and potential capacity of interventions to obtain a desired effect. Both are translated in all occurrences by their closest direct equivalent, the verb *pouvoir* in the indicative mood for *can* and in the conditional mood for *could*, and are therefore not subject to bias in translation. For the auxiliary *may*, which is typically used as a hedging device for mitigation of claims, biased machine-translation by the indicative mood of the verb *pouvoir* (see 1.4. and Example 3 below) accounts for 142/155 (~92%) of its occurrences within a positive lexico-grammatical pattern signaling potential effectiveness of interventions.

**Example 3.** Biased translation of *may* within a positive pattern.

Tocolysis may improve blood flow and therefore improve the baby's well-being.
MT: La tocolyse peut améliorer la circulation sanguine et doncle bien-être du bébé.
[= intervention MAY/CAN improve outcomes]

Translation by the indicative mood of the verb *pouvoir* represents approximately 89% (348/391) of all occurrences of *may* in the corpus sample, while the remaining 11% mainly correspond to the use of the conditional mood. Again, for comparison, on the aforementioned larger corpus of Cochrane abstracts, the indicative mood was present in 54% of human-translated texts versus 45% of texts pre-translated by a statistical engine and post-edited by professionals. This seems to confirm the previous observation on neural MT tending towards human-like output in the translation of modal markers—perhaps even to the point of exaggerating typically human characteristics. Finally, for the modal auxiliary *might*, used for signaling higher uncertainty than *may*, the neutral and functionally equivalent translation choice, the verb *pouvoir* in the conditional mood, is chosen in most cases, biased translation by the indicative mood representing 15% (3/20) of the occurrences of *might* within a positive lexico-grammatical pattern.

Again, in most cases, the biased translation suggested by the MT engine for modal auxiliaries within positive patterns was accepted as such by the student post-editors, even when the shift in the level of certainty was as marked as for the auxiliary *might* translated by the indicative mood (see Table 3). Generally speaking, students tended to modify more frequently the suggested translation for

modal auxiliaries than for evidential verbs in the MT output. Indeed, on a total of seven occurrences in the corpus sample, students edited MT output for the auxiliary *may* from the indicative mood into the conditional mood. In one instance, however, a neutral and functionally equivalent translation of *may* in the MT output was changed into a biased one by the post-editor (i.e., conditional to indicative mood), while in three other instances, the bias in translation was further reinforced at the post-editing stage through other choices (see Example 4 below).

**Example 4.** Bias reinforced at the post-editing stage.

Acupuncture plus routine primary physician care may improve pain and function compared to routine primary physician care alone.

MT: L'acupuncture et les soins médicaux primaires de routine peuvent améliorer la douleur et le fonctionnement comparativement aux soins médicaux primaires de routine seuls.

PE:　L'acupuncture　combinée　à　des　visites　de　routine　chez　le　médecin peuvent se révéler plus efficaces pour réduire la douleur et améliorer le fonctionnement de la hanche que des visites seules.

[= intervention MAY/CAN improve outcomes vs MAY/CAN prove to be more effective]

In the example above, the student post-editor has added a positively biased evidential verb (*se révéler*) which reinforces the bias originating from the use of the indicative mood in the translation of the modal auxiliary *may*.

3.1.3. Modal Clusters

Modal clusters combine different markers, for instance evidential verbs, modal auxiliaries, and adjectives or adverbs, within a larger lexico-grammatical pattern for more precise signaling of the level of proof. Such clusters are occasionally subject to biased translation of the different markers in the MT output (see Example 5a,b below).

**Example 5a.** Biased translation of a modal cluster within a positive lexico-grammatical pattern.

The review of available evidence found that combinations of antipsychotics may be more effective ( … )

MT: L'examen des données probantes disponibles a révélé que les associations d'antipsychotiques peuvent être plus efficaces ( … )

[= review REVEALED that intervention MAY/CAN be more effective]

Contributing to the distortion originating from more affirmative lexical choices for modal markers, another biased translation observed in the MT output in the context of these clusters is the omission of hedges, such as the nominal group *some evidence* in Example 5b below.

**Example 5b.** Biased translation of an evidential verb reinforced by omission.

One well-controlled study shows some evidence of effect of two interventions for childhood apraxia of speech ( … )

MT: Une étude bien contrôlée démontre l'effet de deux interventions sur l'apraxie de la parole chez l'enfant ( … )

[= study DEMONSTRATES/PROVES the effect of interventions]

In previous research [8], biased translation of modal clusters and specifically the elimination of hedges, which leads to markedly more affirmative statements, was typically observed in human translation. Although only a few occurrences (3/103) were observed in the present corpus sample, this

again contributes to the previous observations on too convincing mimicking of human translation by neural MT as a potential source of distortion in the communication of specialized knowledge.

### 3.1.4. Existential Framing Structures

The existential framing structure (see Section 2, pattern e) has a total of 43 occurrences in the corpus sample, 22 of which correspond to positive lexico-grammatical patterns signaling potential benefits of interventions. Although only three instances of biased translation of the existential structure framing positive patterns are observed, it can be noted that they all correspond to markedly more affirmative statements in the MT output than in the source text (see Example 6a,b below). The biased translation combines the omission of a nominal group built around *evidence*, as already observed above for modal clusters, with the adding of an evidential verb signaling a high level of proof such as *prouver* or *démontrer*.

**Example 6a.** Biased translation of an existential framing pattern.

There is evidence that burns treated with honey heal more quickly ( … )
MT: Il est prouvé que les brûlures traitées au miel guérissent plus rapidement ( … )
[= it is PROVEN that intervention is more effective]

**Example 6b.** Biased translation of an existential framing pattern.

There was some evidence that trihexyphenidyl may improve individual goals set by the child and family ( … )
MT: Il a été démontré que le trihexyphénidyle peut améliorer les objectifs individuels fixés par l'enfant et sa famille ( … )
[= it has been DEMONSTRATED/PROVEN that intervention MAY/CAN improve outcome]

In accordance with the previous observations, this very human-like translation combining omission and addition—and resembling something like biased interpretation—could potentially lead to important distortion in the interpretation of research results and thus, the communication of specialized knowledge.

### 3.2. Negative Lexico-Grammatical Patterns Signaling Lack of Evidence of Effect

The search for negative patterns in active or passive voice (see Section 2, patterns f and g) through specific markers yielded a total of 465 results in the corpus, which were then manually filtered to 386 occurrences of the patterns under study. A functional and neutral translation would use a simple form of negation, while the typical biased translation consists in using a stronger form of negation (i.e., *aucun*, see Section 1.4). The negative bias can also occasionally be reinforced by clustering of markers subject to biased translation, for instance evidential verbs or modal adjectives (see Table 4).

**Table 4.** Specific markers subject to biased machine translation within negative patterns.

| Marker | Occurrences within Pattern | >Biased MT | >Biased PE |
|---|---|---|---|
| Negation (no/any) | 386 | 218 (~56%) | 212 |
| NEG + Evidential V | 134 | 13 (~10%) | 12 |
| NEG + modal adjective | 59 | 0 | 0 |

### 3.2.1. Negative Patterns in Active or Passive Voice

The stronger form of negation is used in the translation of approximately 56% (218/386) of the total occurrences of the pattern in the corpus sample. Example 7 below illustrates the typical biased translation, whereas the source text here uses a more neutral form of negation (i.e., *did not show any*

*difference* vs. *showed no difference*), in accordance with the principle of caution regarding any affirmation of lack of effect (see Section 1.4).

**Example 7.** Biased translation of negation markers within a negative pattern.

The trial did not show any difference in effect between those participants given rivastigmine and those given placebo.

MT: L'essai n'a montré aucune différence d'effet entre les participants ayant reçu la rivastigmine et ceux ayantreçu le placebo.

[= trial showed NO (= strong negation) difference in effect]

Previous research on a larger corpus sample of translated Cochrane abstracts [8] suggests the stronger form of negation is slightly more frequent in human-translated texts (~28% of total occurrences of the pattern) than in post-edited statistical MT (~25%). The high incidence of the stronger negation in the present corpus sample could thus be seen as another indicator of a tendency in neural MT output towards exaggeration of typically human-like features in the translation of lexico-grammatical patterns, much like what was observed for the modal auxiliary *may* within positive patterns.

As already observed in the case of positive patterns, students tended to accept biased translation suggestions by the MT engine as such: in only six instances (out of 218), was the strong negation suggested by the MT engine (*aucun*) changed into a more neutral negative form (*ne . . . pas*) at the post-editing stage (see Table 4). Moreover, in another six instances a neutral form of negation in the MT output was actually changed by the student post-editors into a stronger negation. These instances of over-editing of MT output are, however, not included in the present counts, since the focus here is on distortions originating from under-editing of biased NMT output.

3.2.2. Negative Clusters

The negative pattern under study frequently (59/386) contains modal adjectives such as *clear* or *important*, which are in all occurrences machine-translated by their closest direct equivalents and are therefore not subject to bias in translation. Negative clusters are even more frequently formed by associating an evidential verb with a negative pattern (134/386), as in Example 8a,b below. In approximately 10% (13/134) of these occurrences, both the evidential verb and the negative marker are subject to biased translation. As already observed in the case of positive patterns, evidential verbs *find* and *show* are most likely candidates for more affirmative, biased machine-translation (i.e., the verb *révéler*) which contributes to the negative bias originating from the use of the stronger form of negation (see Example 8a below). As discussed in 1.4., a functional translation would associate a more neutral lexical choice for the evidential verb *find* (i.e., *trouver* or *montrer*) with a simple negation (*ne . . . pas*).

**Example 8a.** Biased translation of different markers within a negative pattern

This found no difference in the likelihood of the cancer regrowth between the two groups.

MT: Cette étude n'a révélé aucune différence dans la probabilité de réapparition du cancer entre les deux groupes.

[= study REVEALED NO (= strong negation) difference]

Thus, as already observed for positive patterns, neural MT occasionally produces translations that mimic human translation through the use of more affirmative choices. Such biased choices have the potential to distort readers' interpretation of the intended degree of certainty of the results. This is for instance the case in Example 8b below, where the evidential verb subject to biased translation is markedly more hedged (*suggest*) than the translation proposal by DeepL (*révéler*).

**Example 8b.** Biased translation of different markers within a negative pattern.

Pooled data from two studies (509 participants) <u>suggests no clear difference</u> in risk of wound infection

MT: Les données regroupées de deux études (509 participants) <u>ne révèlent aucune différence claire</u> dans le risque d'infection des plaies

[= data REVEALS NO (= strong negation) difference].

## 4. Discussion

In accordance with the growing body of evidence on NMT, results of this study suggest that NMT output tends to contain textual features very similar to human-translated texts. In the English-French language pair, this tendency is particularly manifest in the more affirmative, i.e., less hedged, presentation of claims, specifically in the context of lexico-grammatical patterns used in the communication of specialized knowledge. In the medical domain, where claims are rarely definitive and hedges abound, these more affirmative translation solutions can potentially distort readers' interpretation of research results and, specifically, the effectiveness of interventions. Distortion due to biased translation is, however, highly context-dependent. Indeed, it must be stressed that most of these translations for specific markers, such as the stronger form of negation or the indicative mood for modal auxiliaries, are extremely frequent and perfectly acceptable in most contexts. The potential for bias therein is only manifest in the specific context of lexico-grammatical patterns used in the communication of medical knowledge. As hedging is used in medical language to convey actual uncertainty, more affirmative translation can result in shifts in the degree of certainty of the authors' claims.

The results also show that student post-editors mostly tend to spontaneously accept MT output as such, and rarely edit these suggestions that convincingly mimic fluent human translation (see Tables 2–4 in Section 3). The distortion in translation can occasionally even be reinforced by human intervention at the post-editing stage. These results are not surprising, given the highly contextual nature of distortion due to biased translation discussed above. Therefore, although the results of the present study regarding post-editing are naturally limited by post-editor profile, it seems plausible to suppose that experienced translators would mostly not spontaneously edit such suggestions either. Indeed, the very potential for distortion in these translation solutions was initially observed by domain experts when reviewing translations produced by professionals (see Section 1.2). Most likely, a medical and statistical background, or specific training in the requirements for the presentation of medical research results, would be necessary to be able to spot and edit most of these potentially biased translation suggestions in the MT output as well. It would nonetheless make for an interesting follow-up study to compare student productions with post-editing by professionals in order to confirm to what extent the observed tendencies are influenced by post-editor profile and experience, or lack thereof. Previous research has indeed suggested that (statistical) MT has a leveling effect in terms of errors in the final output: the difference between professional translators and novices in terms of error counts is less marked in post-editing than in editing fuzzy matches from a translation memory [23] (p. 51). In view of this, and given the relatively high proportion of lexico-grammatical distortions observed in professional human translation in previous work (see Section 1.6), it is thus tempting to hypothesize that professionals would perform rather similarly to students when post-editing NMT suggestions for the translation of lexico-grammatical patterns.

Although the small number of edits performed by the students on biased MT output limits conclusions on any individual differences, some observations can be made regarding the distribution of edits between students and in time. A total of 14 instances of editing biased MT output (see Tables 2–4) were observed in 8 different texts produced by 7 different student post-editors, including one non-native speaker of French. All of these edits were performed on texts produced roughly in the second half of the course, i.e., after the importance of lexico-grammatical patterns in the communication of specialized knowledge and the potential for distortion in their translation had been discussed in

class. Edits are distributed unevenly between students: four students performed a single edit each, while two students performed three edits each, and one student performed a total of four edits on two different texts. Thus, although non-native speakers are less likely to detect the subtle nuances involved in the interpretation of lexico-grammatical patterns, it can be noted that most native speakers did not edit biased MT output either, even after having been made aware of these issues. Finally, it can be noted that although the edits were not correlated with the grades obtained for the course, the highest numbers of edits were performed by two students ranking among the highest grades obtained for the course. The issue of individual preferential differences between students would also make up for a potential follow-up study.

Notwithstanding these limitations, the results of this study thus highlight the need to pay particular attention to lexico-grammatical patterns and modal markers when post-editing NMT. This holds particularly true for languages for specific purposes, where such patterns and markers fulfill essential functions in the realization of the communicative purpose of texts. In addition, these results confirm the need to raise awareness among student post-editors of the importance of lexico-grammatical patterns in the communication of specialized knowledge and of adapting post-editing effort to the text type and genre. Such awareness and adaptation appear paramount for achieving functional equivalence in translation through consideration of the communicative purpose of the text, which is something the machine will not be able to do in the foreseeable future.

**Funding:** This research received no external funding.

**Acknowledgments:** The author would like to thank the reviewers and editors of this special issue for their constructive comments and insightful suggestions during the review process.

**Conflicts of Interest:** The author declares no conflict of interest.

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
