# Peer review of "Post-Editing Neural MT in Medical LSP: Lexico-Grammatical Patterns and Distortion in the Communication of Specialized Knowledge"

_informatics, doi:10.3390/informatics6030026_

Round 1
Reviewer 1 Report
While some of the claims of 'distortion' might be felt by some to be exaggerated (it will depend on the observer), the large majority of the examples are convincing. The article gives valuable further evidence of typical shortcomings of NMT that have been mentioned elsewhere and that post-editors should be aware of. The focus on medical texts and on the wording of medical evidence is definitely pertinent. This is an excellent article in many ways.
I indicated some minor points in the text (see version attached), mostly to do with slightly un-English wording (e.g. "have for function" is not readily used in English).

Author Response
Thank you for your careful review of the paper.
While some of the claims of 'distortion' might be felt by some to be exaggerated (it will depend on the observer), the large majority of the examples are convincing.
RE: I have provided more information in the Introduction section to back up claims of biased translation but also to acknowledge the subjective nature of the interpretation.
I indicated some minor points in the text (see version attached), mostly to do with slightly un-English wording (e.g. "have for function" is not readily used in English).
RE: These have been modified accordingly.
Reviewer 2 Report
Summary
This contribution presents research on biased translations by neural MT (both before and after post-editing) in the English to French translation of medical abstracts. The researcher analyses the DeepL MT and post-editing of master students on medical LSP, focussing on lexico-grammatical patterns expressing positive or negative situations and which may cause biased translations. They conclude that NMT uses and even abuses human-like translation strategies. In this case, this manifests in the "more affirmative" description of claims.
This paper is written clearly and in good English. It is well-structured and addresses an interesting question that deserves further attention. Overall, the methodology is valid and scrupulously described and this paper could be a good contribution to the research field.
Remarks
However, a few problems need to be addressed, mainly the definition of "biased" translations. As the main topic of this paper, it deserves perhaps more attention in the introduction, especially since some of the so-called "biased" translations (e.g. 5a, 7, 8a), seem perfectly fine with me. Some of the examples are very clear (e.g. 6a and 6b) and well-chosen to illustrate the issues, but in others, such as example 7, there is no clear "bias" to me. Either the theoretical framework about such biased translations needs to be elaborated, so that it is clear why such seemingly correct translations are biased, or the unclear examples should be removed and only the clearly biased translations should be discussed. It could also help to always provide an unbiased translation option for comparison, not just in the description of the different patterns, but in the specific examples. Another strategy would be to compare the MT and post-editing to human translation, which would be especially interesting in the cases where the bias is unclear.
It also seems important to know if the students had any experience with post-editing and Memsource and if this could have potentially influenced the results. Perhaps reference could be made to research looking into the difference between students and professional translators to nuance the results and make a hypothesis about the comparative performance of professionals versus students in this task.
Minor remarks
Instead of always referring to the PhD research with a footnote, it would be more readable to do this in the text itself.
Line 104: which study?
Line 138: rewrite first sentence more clearly
Line 154-159: it would be useful to know exactly which markers were assessed with which procedure (e.g. by referring to the letters in the above description)
Line 349 ("Clusters are (...)"): this sentence could be clarified with an example
Author Response
Thank you very much for your insightful comments.
However, a few problems need to be addressed, mainly the definition of "biased" translations. As the main topic of this paper, it deserves perhaps more attention in the introduction, especially since some of the so-called "biased" translations (e.g. 5a, 7, 8a), seem perfectly fine with me. Some of the examples are very clear (e.g. 6a and 6b) and well-chosen to illustrate the issues, but in others, such as example 7, there is no clear "bias" to me. Either the theoretical framework about such biased translations needs to be elaborated, so that it is clear why such seemingly correct translations are biased, or the unclear examples should be removed and only the clearly biased translations should be discussed.
RE: More information has been provided in the Introduction section to back up claims of potential bias, but also to acknowledge the subjective nature of the interpretation.
It could also help to always provide an unbiased translation option for comparison, not just in the description of the different patterns, but in the specific examples.
RE: For text economy, unbiased translations of the entire examples have not been added, but for all the markers subject to biased translation in the examples, such options have been discussed.
Another strategy would be to compare the MT and post-editing to human translation, which would be especially interesting in the cases where the bias is unclear.
RE: For these examples, no HT version is available for comparison purposes, as the corpus is issued from an actual production context using only PEMT. However, this would certainly make for an interesting follow-up study.
It also seems important to know if the students had any experience with post-editing and Memsource and if this could have potentially influenced the results.
RE: Information has been added accordingly.
Perhaps reference could be made to research looking into the difference between students and professional translators to nuance the results and make a hypothesis about the comparative performance of professionals versus students in this task.
RE: Thank you very much for this suggestion, this has been added to the Discussion section.
Minor remarks
- Instead of always referring to the PhD research with a footnote, it would be more readable to do this in the text itself. >> References changed accordingly.
- Line 104: which study? >> Clarification added.
- Line 138: rewrite first sentence more clearly >> Split into two sentences for more clarity.
- Line 154-159: it would be useful to know exactly which markers were assessed with which procedure (e.g. by referring to the letters in the above description) >> Corresponding letters added.
- Line 349 ("Clusters are (...)"): this sentence could be clarified with an example >> Clarification added (the examples below illustrate this phenomenon).
Reviewer 3 Report
The paper presents a study investigating specific English lexico-grammatical patterns and their French translations in both neural MT output and in post-edited versions of that output. The methodology appears overall sound, and the results present some interesting observations regarding both patterns in the NMT output and in the PE versions, which is likely to be of interest to other researchers working on this topic. The paper is generally well-written and clear, but there are some specific points where further clarification would be necessary.
The main potential weakness is that the definition of "biased" translation, which is central to the overall argument. How was a specific word choice (or structure) in the target determined to be more/less strong than the source text? What was the basis used for this determination? It is noted in the paper that the selection of these specific patterns and the methodology in general is based on ongoing work and previously published articles by the author, but it would be helpful for readers of this specific paper to add a more detailed description of the methodology. If the determination was (presumably?) made by the author alone, that would also seem to introduce some subjectivity into the assessment, which should be acknowledged more clearly. While some of the examples given seem rather straightforward (e.g. example 5b with the omission), the distinction of certainty made by the author is not always self-evident, particularly if the reader is not a French-speaker. The glosses provided are helpful, but it is not necessarily obvious that "demonstrate", for example, is stronger than "show". In some cases, e.g. in the discussion of the modals "may"/"might" and indicative vs conditional mood, it would be good to offer a clear explanation of what the author would consider a non-biased translation, and why. I do not necessarily disagree with the author's assessments, but they should be explained and justified more explicitly.
A second point that should be clarified is the set-up for collecting the PE data. From the number of students and the number of texts I assume that different students edited different texts, but this could be made more explicit (how many texts per student etc.). Since the overall conditions may also affect the end result, they should also be clarified. What were the instructions given to the student post-editors? What resources (e.g. dictionaries) did they have at their disposal? Did they have a time limit or were they able to work at their own pace? Since the post-editors in this experiment were students instead of professionals, and presumably carried out the post-editing as course assignment/experiment, it might be good to also discuss the effect that may have.
Previous research has observed that different post-editors can produce very different PE versions even of the same text based on apparent individual preferences (the point is mentioned e.g. in the Koponen et al. paper cited, and Giselle de Almeida's PhD thesis from 2015 discusses such observations in more detail). Considering this, it would be interesting to know whether any differing patterns can be observed between the students in this study, and whether one or more of them may be particularly prone to contributing a notable share of specific types of cases.
Some more detailed comments on specific points in the text:
p. 1
"... seem to convincingly mimic human translation processes." -> Agreed that NMT *output* may mimic a *product* of human translation; however, it does not follow that an NMT system mimics the *process* of a human translator. This is perhaps a minor point, but the distinction of product and process is rather significant in translation studies.
p. 2
"The focus of this paper is on the translation of lexico-grammatical patterns...." -> The term "lexico-grammatical pattern" as such is quite vague and broad, and it would be easier for the reader to follow the discussion here if a concrete example of the type of pattern being investigated was given here early on.
"Thus, translating such lexico-grammatical patterns ... is likely to result in a shift ..." -> What is meant by "likely"? How commonly does this happen? Although a reference to the previous study is provided, giving more detail about how this was studied, how common such an occurrence was and how representative the study in question is would contextualize the issue more clearly.
p. 3
"The following lexico-grammatical patterns and specific markers therein were selected..." -> Why these patterns? What were the observations and findings in the previous studies that led to these specific patterns being selected?
p. 4
"...patterns that have for function to express..." -> This sentence is quite difficult to parse, consider rephrasing.
p. 6
"Four of these modal verbs can occur in the context of the positive lexico-grammatical pattern..." -> Is there some particular reason why the other four *cannot* occur in that pattern? (Or is the case simply that they do not happen to occur in this specific dataset?)
"... indicative mood of the same verb (see Introduction and Example 3 below)..." -> As a reader I found this confusing - why not just state directly which verb?
"Translation by the indicative mood of the verb pouvoir represents approximately 89%..." -> Since this is raised as a key example: What are the other translations that occur? Which of them (if any?) are considered non-biased and why?
p. 7
"Although only a few occurrences (3) were observed..." -> It is difficult to assess how meaningful this observation is without knowing how many similar cases there were (are we dealing with 3 out of 10 such sentences? 3 out of 100?).
p. 9
"...neural MT occasionally produces translations that mimic to perfection the tendency of human translators..." -> "Perfection" seems to be somewhat overstating the case, perhaps.
p. 10
"NMT tends to use ... human-like translation strategies" -> Somewhat related to my previous comment re. process: "Translation strategies" would seem to imply approaches that a human translator (more or less) consciously and often (more or less) consistently adopts for a given text and situation. Although the observations regarding how often NMT output contains textual features similar to what appear in a translation produced by human choosing a specific strategy are interesting, I am not convinced it is possible (or necessary) to ascribe such strategies to the NMT system.
Author Response
Thank you for these very useful suggestions for revision.
The main potential weakness is that the definition of "biased" translation, which is central to the overall argument. How was a specific word choice (or structure) in the target determined to be more/less strong than the source text? What was the basis used for this determination? It is noted in the paper that the selection of these specific patterns and the methodology in general is based on ongoing work and previously published articles by the author, but it would be helpful for readers of this specific paper to add a more detailed description of the methodology. If the determination was (presumably?) made by the author alone, that would also seem to introduce some subjectivity into the assessment, which should be acknowledged more clearly. While some of the examples given seem rather straightforward (e.g. example 5b with the omission), the distinction of certainty made by the author is not always self-evident, particularly if the reader is not a French-speaker. The glosses provided are helpful, but it is not necessarily obvious that "demonstrate", for example, is stronger than "show". In some cases, e.g. in the discussion of the modals "may"/"might" and indicative vs conditional mood, it would be good to offer a clear explanation of what the author would consider a non-biased translation, and why. I do not necessarily disagree with the author's assessments, but they should be explained and justified more explicitly.
RE: More information has been provided in the Introduction section to develop and back up claims of potential bias, but also to acknowledge the subjective nature of the interpretation.
A second point that should be clarified is the set-up for collecting the PE data. From the number of students and the number of texts I assume that different students edited different texts, but this could be made more explicit (how many texts per student etc.). Since the overall conditions may also affect the end result, they should also be clarified. What were the instructions given to the student post-editors? What resources (e.g. dictionaries) did they have at their disposal? Did they have a time limit or were they able to work at their own pace? Since the post-editors in this experiment were students instead of professionals, and presumably carried out the post-editing as course assignment/experiment, it might be good to also discuss the effect that may have.
RE: Information on the set-up has been added in the Methods section, and comparison with professionals added as potential follow-up in the Discussion section.
Previous research has observed that different post-editors can produce very different PE versions even of the same text based on apparent individual preferences (the point is mentioned e.g. in the Koponen et al. paper cited, and Giselle de Almeida's PhD thesis from 2015 discusses such observations in more detail). Considering this, it would be interesting to know whether any differing patterns can be observed between the students in this study, and whether one or more of them may be particularly prone to contributing a notable share of specific types of cases.
RE: Thank you for these references. The compiled corpus designed for statistical study does not, as such, allow for study of individual differences between students. However, generally speaking, very few changes were made to the translation of the patterns under study (Tables 2 a 3).
Some more detailed comments on specific points in the text:
p. 1. "... seem to convincingly mimic human translation processes." -> Agreed that NMT *output* may mimic a *product* of human translation; however, it does not follow that an NMT system mimics the *process* of a human translator. This is perhaps a minor point, but the distinction of product and process is rather significant in translation studies.
RE: Thank you for highlighting this important issue, changes have been made accordingly across the paper.
p. 2. "The focus of this paper is on the translation of lexico-grammatical patterns...." -> The term "lexico-grammatical pattern" as such is quite vague and broad, and it would be easier for the reader to follow the discussion here if a concrete example of the type of pattern being investigated was given here early on.
"Thus, translating such lexico-grammatical patterns ... is likely to result in a shift ..." -> What is meant by "likely"? How commonly does this happen? Although a reference to the previous study is provided, giving more detail about how this was studied, how common such an occurrence was and how representative the study in question is would contextualize the issue more clearly.
RE: The Introduction section has been developed for more precise description of LG bias and its potential for distortion.
p. 3. "The following lexico-grammatical patterns and specific markers therein were selected..." -> Why these patterns? What were the observations and findings in the previous studies that led to these specific patterns being selected?
RE: Clarification added.
p. 4. "...patterns that have for function to express..." -> This sentence is quite difficult to parse, consider rephrasing.
RE: Rephrased accordingly.
p. 6. "Four of these modal verbs can occur in the context of the positive lexico-grammatical pattern..." -> Is there some particular reason why the other four *cannot* occur in that pattern? (Or is the case simply that they do not happen to occur in this specific dataset?)
"... indicative mood of the same verb (see Introduction and Example 3 below)..." -> As a reader I found this confusing - why not just state directly which verb?
"Translation by the indicative mood of the verb pouvoir represents approximately 89%..." -> Since this is raised as a key example: What are the other translations that occur? Which of them (if any?) are considered non-biased and why?
RE: Clarification added on these three points.
p. 7. "Although only a few occurrences (3) were observed..." -> It is difficult to assess how meaningful this observation is without knowing how many similar cases there were (are we dealing with 3 out of 10 such sentences? 3 out of 100?).
RE: Clarification added.
p. 9. "...neural MT occasionally produces translations that mimic to perfection the tendency of human translators..." -> "Perfection" seems to be somewhat overstating the case, perhaps.
RE. Indeed, removed and rephrased.
p. 10. "NMT tends to use ... human-like translation strategies" -> Somewhat related to my previous comment re. process: "Translation strategies" would seem to imply approaches that a human translator (more or less) consciously and often (more or less) consistently adopts for a given text and situation. Although the observations regarding how often NMT output contains textual features similar to what appear in a translation produced by human choosing a specific strategy are interesting, I am not convinced it is possible (or necessary) to ascribe such strategies to the NMT system.
RE: Again, thank you very much for highlighting this important issue, changes have been made accordingly across the paper.
Round 2
Reviewer 1 Report
I am happy that the (relatively) subjective nature of judgements on "distortion" has now been admitted explicitly and that it has been clarified that the judgements used in the paper are based on majority opinion among domain specialists.
I understand that this required an explanation of strategies and working methods used in the underlying PhD research. This account is now probably somewhat lengthier than necessary (but maybe requested by another editor) - at any rate the general context is now entirely clear.
Author Response
Thank you again for your careful review of the paper.I understand that this required an explanation of strategies and working methods used in the underlying PhD research. This account is now probably somewhat lengthier than necessary (but maybe requested by another editor) - at any rate the general context is now entirely clear.
I agree that the account of previous research in the Introduction section is now somewhat lengthy. It seemed necessary to give all this background information to back up claims on distortion, given its subjective and contextual nature. If deemed necessary, I will of course be happy to shorten this section as appropriate.
Reviewer 2 Report
The original concerns have been sufficiently addressed in this reviewed version. The bias in the translations is better explained and justified with clearer references to the PhD research. The writing has been improved as well, including corrections of the minor remarks in the original review. This new version is a definite improvement over the first version.
However, one of the improvements also raised new questions. As requested, the profile of the participants has been elaborated. Considering the difficulty and cost of finding participants for such a study, it seems justified to work with 2nd year master’s students, provided the conclusions are nuanced accordingly. Nevertheless, due to the focus of this study on extremely subtle distinctions in the translations (despite the improved explanations, the issues remain undeniably subtle and difficult to detect), I was very surprised to find that “students had (…) diverse language combinations, with 8 non-native speakers of French in the class.” To have inexperienced students rather than experienced professionals perform the post-editing is enough of a complication, but to also work with students whose language combination is not even the same as the texts, and even students whose native language is not the target language endangers the validity of the entire experiment. With this in mind, it seems no more than logical that hardly any of the biased MT output is post-edited, since the issues discussed would be difficult enough to detect for experienced post-editors; it is simply too much to expect from students who lack both experience ánd knowledge of the language combination. Results from students who did study the relevant language combination might still be worth discussing (with the necessary caution), but I don’t think any valid conclusions can be drawn from the others with this methodology, at least not without proof that their overall post-editing was of a sufficient quality to indicate that they grasp all the nuances and intricacies of both source and target language well enough to detect the bias discussed (which I highly doubt).
In a related note, I still would like to know more about the impact of the students’ experience. It is now stated that “most were novices to both post-editing and medical translation at the beginning of the class. Experience in both was acquired over the duration of the class, and theoretical teachings and practical exercises were used in support of the post-editing work done by students.” From the text, I assume the study was performed rather in the beginning of the class, so before the students learnt more about these skills(?). Was biased MT specifically discussed in class, before the study was performed? Was this a graded exercise? Since there was a wide age range, were there also students with prior experience? And most importantly: was there any notable effect of experience, i.e. were students with more experience/higher marks during the class/the appropriate language combination more likely to spot and correct biased MT?
In conclusion, the text was satisfactorily improved based on the first review, but the added information regarding the students’ profiles raised important concerns regarding the methodological validity. Irrelevant results should be removed from the study and conclusions regarding the post-editing should be extremely nuanced. Since much of the focus of this research is on the MT itself, that part of the study can still be presented in its current form. Any results and conclusions regarding the post-editing, however, should be critically evaluated and removed when necessary.
Author Response
Thank you again for your careful review of the paper.
Considering the difficulty and cost of finding participants for such a study, it seems justified to work with 2nd year master’s students, provided the conclusions are nuanced accordingly.
Additional clarification has been added in the Methods section on the fact that this is a corpus study on data derived from an actual production context and not by any means a controlled experiment. Study limitations have been discussed in regard to the results on post-editing.
Nevertheless, due to the focus of this study on extremely subtle distinctions in the translations (despite the improved explanations, the issues remain undeniably subtle and difficult to detect), I was very surprised to find that “students had (…) diverse language combinations, with 8 non-native speakers of French in the class.” To have inexperienced students rather than experienced professionals perform the post-editing is enough of a complication, but to also work with students whose language combination is not even the same as the texts, and even students whose native language is not the target language endangers the validity of the entire experiment.
Thank you for raising this issue. The wording regarding student language combinations was unfortunately extremely ambiguous and misleading in the previous version. All students, non-native speakers included, had been tested in the EN-FR language pair in order to be admitted to the program and this was the main language combination used for most courses. This has been clarified in the Methods section. Although the main focus of the corpus study is to explore NMT output in the translation of LG patterns, general observations are made on student performance and it is not suggested that their results can be applied to professional post-editors.
With this in mind, it seems no more than logical that hardly any of the biased MT output is post-edited, since the issues discussed would be difficult enough to detect for experienced post-editors; it is simply too much to expect from students who lack both experience and knowledge of the language combination. Results from students who did study the relevant language combination might still be worth discussing (with the necessary caution), but I don’t think any valid conclusions can be drawn from the others with this methodology, at least not without proof that their overall post-editing was of a sufficient quality to indicate that they grasp all the nuances and intricacies of both source and target language well enough to detect the bias discussed (which I highly doubt).
The issue of language combination has been clarified in the Methods section, and some detail added on student profiles (i.e. professional master with selection of students based on testing). Moreover, the highly context-dependent nature of bias has been clarified in the Discussion section. It is hypothesized that experienced translators would likely under-edit these suggestions as well, as detection of potential bias as defined by the domain experts would require, in the least, training in the presentation of medical results.
In a related note, I still would like to know more about the impact of the students’ experience. It is now stated that “most were novices to both post-editing and medical translation at the beginning of the class. Experience in both was acquired over the duration of the class, and theoretical teachings and practical exercises were used in support of the post-editing work done by students.” From the text, I assume the study was performed rather in the beginning of the class, so before the students learnt more about these skills(?). Was biased MT specifically discussed in class, before the study was performed? Was this a graded exercise? Since there was a wide age range, were there also students with prior experience? And most importantly: was there any notable effect of experience, i.e. were students with more experience/higher marks during the class/the appropriate language combination more likely to spot and correct biased MT?
Given the small number of edits performed by the students, only tentative conclusions on individual differences can be drawn. Clarification has been added on these issues in the Discussion section.
Reviewer 3 Report
The clarifications and additional information provided address the issues raised in the previous review. The question of individual preferential differences between post-editors might be interesting for some further study. Overall, the language is also good - splitting some of the very long paragraps e.g. on pages 2 and 3 might improve readability, but that is a minor point and not essential. My thanks to the author for the hard work and I look forward to seeing this interesting article in print.
Author Response
Thank you again for your careful review of the paper.
The question of individual preferential differences between post-editors might be interesting for some further study.
Indeed, thank you. Some tentative observations on individual differences have also been added in the Discussion section.
Overall, the language is also good - splitting some of the very long paragraps e.g. on pages 2 and 3 might improve readability, but that is a minor point and not essential.
These have been edited accordingly, and sub-section titles added where appropriate.
Round 3
Reviewer 2 Report
The current version of this paper is much clearer than previous versions and can, in my opinion, be published in its present form.
My main concerns regarding the ambiguity of the "errors" and the experience of the students and its impact have been addressed, explained, and nuanced. As was the case from the beginning, the text is well-written and clearly structured. While the conclusions need to remain tentative due to the small sample size and methodology, this research has a sound methodology and provides valuable insights into bias in both human and machine translation. I hope it can inspire more elaborate research into this relatively unknown subject.